# Sex-dependent, lateralized engagement of anterior insular cortex inputs to the dorsolateral striatum in binge alcohol drinking

**David L Haggerty[1], Brady K Atwood[1,2]***

[1]Department of Pharmacology & Toxicology, Indiana University School of Medicine, Indianapolis, United States; [2]Stark Neurosciences Research Institute, Indiana University School of Medicine, Indianapolis, United States

**\*For correspondence:**
bkatwood@iu.edu

**Competing interest:** The authors declare that no competing interests exist.

**Abstract** How does alcohol consumption alter synaptic transmission across time, and do these alcohol-induced neuroadaptations occur similarly in both male and female mice? Previously we identified that anterior insular cortex (AIC) projections to the dorsolateral striatum (DLS) are uniquely sensitive to alcohol-induced neuroadaptations in male, but not female mice, and play a role in governing binge alcohol consumption in male mice (Haggerty et al., 2022). Here, by using high-resolution behavior data paired with in-vivo fiber photometry, we show how similar levels of alcohol intake are achieved via different behavioral strategies across sexes, and how inter-drinking session thirst states predict future alcohol intakes in females, but not males. Furthermore, we show how presynaptic calcium activity recorded from AIC synaptic inputs in the DLS across 3 weeks of water consumption followed by 3 weeks of binge alcohol consumption changes across, fluid, time, sex, and brain circuit lateralization. By time-locking presynaptic calcium activity from AIC inputs to the DLS to peri-initiation of drinking events we also show that AIC inputs into the left DLS robustly encode binge alcohol intake behaviors relative to water consumption. These findings suggest a fluid-, sex-, and lateralization-dependent role for the engagement of AIC inputs into the DLS that encode binge alcohol consumption behaviors and further contextualize alcohol-induced neuroadaptations at AIC inputs to the DLS.

## eLife assessment

This **valuable** manuscript describes evidence of sex differences in specific corticostriatal projections during alcohol consumption, and this is noteworthy given the increasing rates/levels of drinking in females and their liability for Alcohol Use disorder. The authors provide **solid** evidence of the lateralisation of the activity of the circuit, but other evidence is **incomplete**, particularly with regard to how the drinking measure relates to intoxication. There are some inconsistencies that make it difficult to reconcile the photometry and behavioral data. The findings would benefit from causal assessment in the future. The findings will be of interest to researchers investigating functional circuitry underlying alcohol-driven behaviors.

## Introduction

Binge alcohol consumption, defined as consuming 4–5 or more drinks in about 2 hr for females and males, respectively, represents a large proportion of the deaths associated with problematic alcohol use and is one of the strongest risk factors for developing alcohol use disorder (AUD) (***Mokdad et al.,***

*2004*; *Addolorato et al., 2018*; *Kanny et al., 2018*). Binge drinking is particularly prevalent among young adults, especially among females who have been increasing their alcohol use rates year after year for the past half-decade that now are similar to or outpacing male use rates (*Keyes et al., 2019*; *White, 2020*).

Modeling binge drinking in rodent animal models is largely accomplished through the behavioral paradigm, Drinking in the Dark (DID), in which mice have ad libitum access to alcohol (20% v/v) in their home cage for 2 hr sessions, 3 hr into their dark-light cycle for 3–4 days a week, and for one 4 hr 'binge' session for a single day, followed by 2 or 3 days of abstinence from alcohol (*Rhodes et al., 2005*; *Thiele and Navarro, 2014*). This procedure has great construct and predictive validity for human binge drinking. Mice display similar drinking patterns to humans like front-loading behavior wherein they consume larger amounts of alcohol early in the task compared to later in the task (*Ardinger et al., 2022*). DID also successfully induces pharmacologically relevant levels of blood alcohol content (BAC) in mice that is equivalent to the legal intoxication limit in humans (*Barkley-Levenson and Crabbe, 2014*; *Huynh et al., 2019*).

It is also commonly observed that female mice tend to drink more than males across a wide range of alcohol consumption models, including DID (*Blednov et al., 2005*; *Hwa et al., 2011*; *Satta et al., 2018*; *Sneddon et al., 2019*). Possible reasons are that females tend to drink more than males is due to an increased sensitivity to the rewarding and decreased sensitivity to the aversive properties of alcohol, different pharmacokinetic and pharmacodynamic properties of alcohol, and differences in sex hormone expression levels, such as estrogen which fluctuates across the menstrual cycle (*Czerniak, 2001*; *Overstreet et al., 2004*; *Schramm-Sapyta et al., 2009*; *Finn et al., 2018*; *Sneddon et al., 2019*; *Vandegrift et al., 2020*; *Elvig et al., 2021*). Yet, there are also reported instances in which male and female mice consume identical amounts of alcohol in these drinking models, such as for mice genetically selected to consume high binge levels of alcohol (*Savarese et al., 2021*). And, while it is known that estrogen plays a role in increasing alcohol intake in some models, there is evidence to suggest there is no effect of the estrous cycle on alcohol intake specifically during DID (*Satta et al., 2018*).

Nonetheless, when female sex-dependent differences in alcohol intake are present during DID, they are often solely reported in comparison to males and/or other experimental controls (e.g. ovariectomized mice). In some cases, these measures of quantitative intake are also reported in combination with how BACs differ by sex, but ultimately these two metrics are often the only two reported measures of how animals consume alcohol differently by sex. Thus, there remains very little knowledge on the behavioral strategies used to achieve alcohol intake across the sexes.

Additionally, the neural circuitry that underlies binge drinking remains poorly understood, and descriptions of sex differences for neural circuits currently implicated in binge alcohol consumption remain even more elusive. Of the neural circuits implicated in binge drinking, only a few of these studies to date have included sex as a factor in their analyses (*Rinker et al., 2017*; *Ferguson et al., 2019*; *Siciliano et al., 2019*; *Burnham et al., 2021*; *Dornellas et al., 2021*; *Levine et al., 2021*; *Haggerty et al., 2022*).

Previous work from our laboratory that investigates such circuitry shows that alcohol uniquely alters AIC inputs into the DLS via synaptic alterations in mu opioid and ionotropic glutamate receptors in male mice (*Muñoz et al., 2018*; *Haggerty et al., 2022*). Female mice displayed no alcohol-induced changes in ionotropic glutamate receptor function. Optogenetically activating these AIC inputs to DLS inputs during DID also reduces how much alcohol male mice binge consume.

To further detail the sex differences displayed between male and female mice in DID and build on our understanding of how AIC inputs into the DLS are altered by binge alcohol exposure across time, fluid, and sex, we measured detailed drinking behaviors and presynaptic calcium activity at AIC inputs into the DLS for 3 weeks of water drinking followed by 3 weeks of binge alcohol drinking. Here, we show that males and females consume water similarly, both in amount and behavioral strategy. Next, we described how the drinking mechanics and patterns differ between females and males who binge similar levels of alcohol, such that female mice consume alcohol more efficiently. Also, by measuring inter-DID session water intakes, we show how female, but not male, thirst states predict future alcohol intake. Finally, using intersectional genetic manipulations, we expressed a genetically encoded calcium indicator (GCaMP), bilaterally and exclusively in AIC inputs to the DLS to measure how presynaptic AIC input activity changes across time, sex, fluid, and brain circuit lateralization. By

recording GCaMP activity from AIC inputs bilaterally in the DLS, we show that the AIC inputs into the left DLS strongly encode binge alcohol drinking in males. Furthermore, an increasing history of alcohol binge drinking alters calcium dynamics in male left AIC inputs into the DLS, suggesting decreased left AIC input engagement as a function of increased drinking history. Together, we establish detailed sex-differences in the behavioral strategies females use to binge consume similar amounts of alcohol compared to males. We describe how AIC inputs to the left DLS encode binge alcohol drinking and how the engagement of these inputs changes over the course of binge drinking history across sex, time, and circuit lateralization. Together these findings represent targets for future therapeutic modalities such as transcranial magnetic stimulation (TMS) and focused ultrasound (FUS) that seek to leverage the unique alcohol-induced changes to AIC inputs into the DLS to create more targeted, efficacious treatment approaches for those experiencing AUD.

## Results

### Sex-dependent differences in water and alcohol drinking in the dark

We isolated AIC→DLS circuitry by injecting an anterograde adeno-associated virus (AAV) into the AIC encoding a cre-dependent GCaMP and a retrograde AAV to express cre, a recombinase enzyme, in the DLS and implanted bilateral optic cannulas in the DLS to visualize and record calcium activity in synaptic inputs that arise from AIC projections to the DLS in both male and female mice (*Figure 1a*). For all viral and fiber placements please see *Figure 1—figure supplement 1*.

Animals then underwent 6 weeks of the drinking in the dark (DID) paradigm where they had 0, 2, or 4 hr of access to water each day for 3 weeks, and then were transitioned to alcohol access for an additional 3 weeks (*Figure 1b*). For DID sessions in the first 3 weeks, animals achieved stable levels of water intake across 15 water DID sessions, and there were no sex differences between male and female water drinking (*Figure 1c–d*). By averaging water intakes by week, a clearer understanding of weekly water intakes can be computed with no noted water intake differences across the week or by sex (*Figure 1e–f*). During drinking weeks 4–6, when animals were transitioned from water to alcohol intake, mice reliably consumed binge-like levels of alcohol during the 4 hr sessions, which correlated with pharmacologically relevant levels of intake as based on previous experiments in our laboratory using identical drinking measures (*Figure 1g*; *Haggerty et al., 2022*). We failed to find differences in alcohol intake between males and females across all DID sessions in this paradigm (*Figure 1h*). Nonetheless, when analyzing alcohol consumption by week, for both sexes, there was a greater amount of alcohol drank during week 4 in comparison to week 5 (*Figure 1i*). This could be due to the transition from water to alcohol, in which the total fluid volume consumed during water sessions is larger than alcohol sessions and may be supported by the decreasing levels of alcohol intake seen in early alcohol DID sessions 16–19 as animals adapt to consuming a new fluid type (*Figure 1h*). By week, both male and female animals experienced this decrease in alcohol intake in weeks 5 and 6 as compared to week 4 of DID, but there were no sex differences or interactions between sex and drinking week (*Figure 1j*). Ultimately, both sexes consume binge-like levels of alcohol, and in relation to the measured calcium activity at AIC inputs into the DLS presented below, one can be confident that differences in these signals are not a function of the total alcohol intake by sex.

The amount of water intake between DID sessions was also measured daily using the home cage water bottle. Interestingly, females consumed more water between both water and alcohol DID sessions across all 6 weeks compared to males (*Figure 1k*). Previous inter-session DID water intakes were also correlated to the next DID session intake to see if the amount of water consumed before any water DID session had any influence on the amount of water consumed during the upcoming DID session. We found a modest, yet statistically significant negative correlation (*Figure 1l*). This suggests that increased water intake, a proxy measurement for a quenched thirst state, being achieved before a water DID session can predict lower levels of water consumed during the DID session itself. We found no sex differences in this correlation between males and females (*Figure 1m*). Similarly, we questioned if inter-session water intakes influenced future alcohol DID intakes, which we did not find to be true when looking at the data blind to sex (*Figure 1n*). Yet, the patterning between males and females for thirst states looks strikingly different. While we did not find a statistically significant difference using sex as a covariate in a co-variate analysis of variance (ANCOVA) to determine if the slopes of the regression fit were statistically different, if one a priori separates sex and then runs a

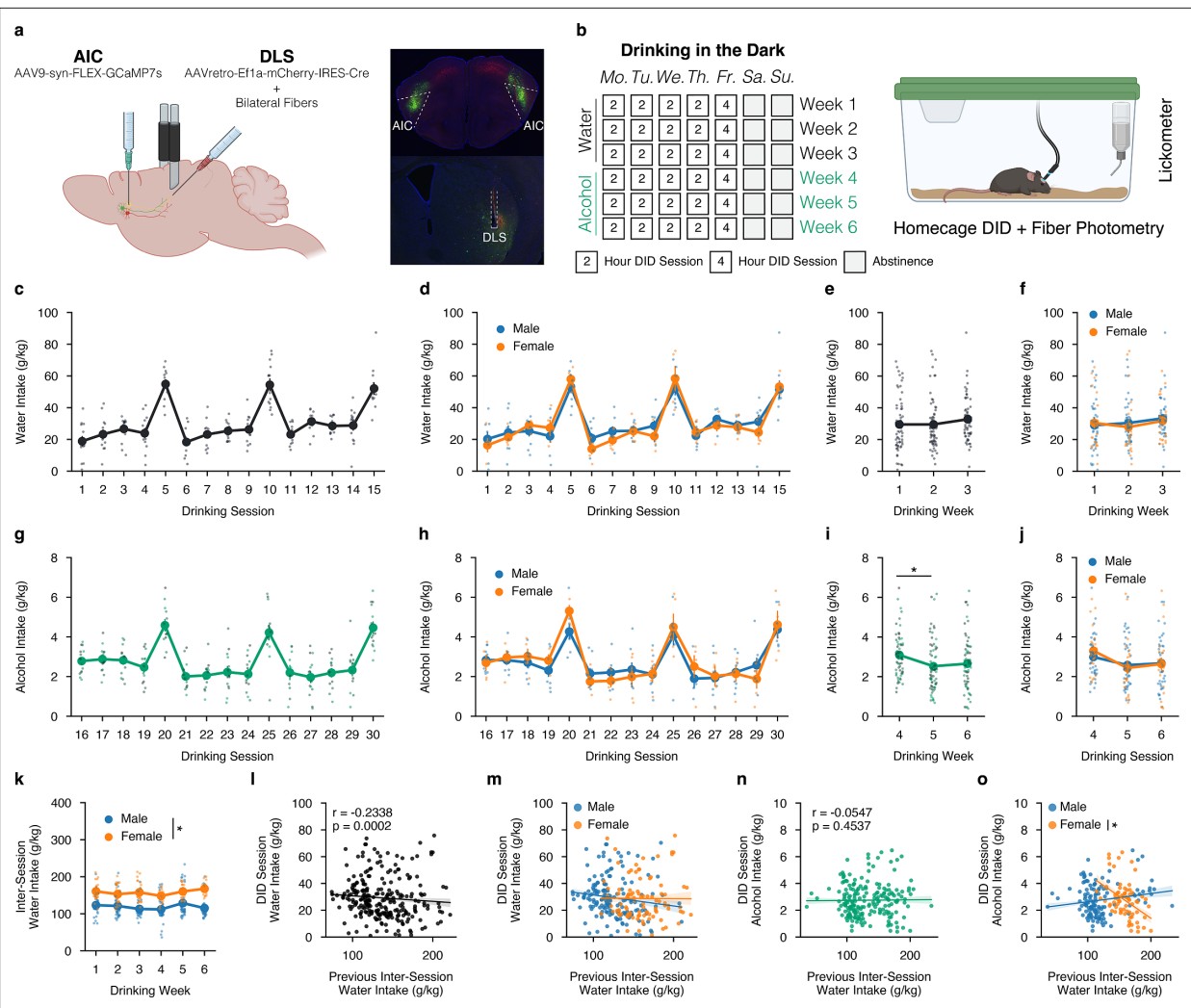

**Figure 1.** Sex-dependent differences in water and alcohol drinking in the dark (DID). (**a**) Representation of viral strategy and expression to record presynaptic calcium activity of anterior insular cortex (AIC) inputs within the dorsolateral striatum (DLS). (**b**) Schematic of drinking in the dark (DID) protocol. (**c**) Group and individual animals' water consumption by DID session (n=14 animals). (**d**) Group and individual animals' water consumption by DID session and by sex (mixed ANOVA, Sex $F_{(1,12)}$ = 0.1032, p=0.7535; male n=9, female n=5). (**e**) Group and individual animals' water consumption by DID week (rmANOVA, Week $F_{(2,26)}$ = 2.4712, p=0.1041). (**f**) Group and individual animals' water consumption by DID week and by sex (mixed ANOVA, Sex $F_{(1,7)}$ = 0.1032, p=0.7535; male n=9, female n=5). (**g**) Group and individual animals' alcohol consumption by DID session (n=14 animals). (**h**) Group and individual animals' alcohol consumption by DID session and by sex (mixed ANOVA, Sex $F_{(1,7)}$ = 0.4714, p=0.5144; male n=9, female n=5). (**i**) Group and individual animals' alcohol consumption by DID week (rmANOVA, Week $F_{(2,26)}$ = 7.8861, p=0.002, $np^2$=0.0992, Power (1 - β)=0.9999; Week 4 to Week 5, p=0.0012; male n=9, female n=5). (**j**) Group and individual animals' alcohol consumption by DID week and by sex (mixed ANOVA, Sex $F_{(1,7)}$ = 0.4714, p=0.5144; male n=9, female n=5). (**k**) Group and individual animals' inter-session water consumption by DID week and by sex (mixed ANOVA, Sex x Drinking Week $F_{(5,60)}$ = 2.411, p=0.0466, $np^2$=0.1673, Power (1 - β)=1.0; male n=9, female n=5). (**l**) Previous inter-session water intakes negatively correlate with the next DID session water intake (Shepherd's pi correlation, r=–0.2383, p=0.0002). (**m**) Previous inter-session water intakes do not correlate with next DID session water intake by sex (ANCOVA, Sex F(1,257) = 0.3048, p=0.5814; male n=9, female n=5). (**n**) Previous inter-session water intakes do not correlate with the next DID session alcohol intake (Shepherd's pi correlation, r=–0.0547, p=0.4537). (**o**) Previous inter-session water intakes do not correlate with next DID session alcohol intake by sex (ANCOVA, Sex $F_{(1,315)}$ = 0.0101, p=0.9201; male n=9, female n=5), but when a priori splitting by sex females, but not males, future alcohol intakes are negatively correlated with increasing amounts of inter-session water intakes (males only r=0.1554, p=0.0916; females only r=–0.4133, p=0.0007, Power (1 - β)=0.9328).

The online version of this article includes the following figure supplement(s) for figure 1:

**Figure supplement 1.** Locations of anterior insular cortex (AIC) injections and dorsolateral striatum (DLS) injections and fiber optic cannula placements.

regression, the correlation for male intersession water intakes predicting alcohol DID intakes is not significant (p=0.0916), whereas increasing female intersession water intake is negatively and significantly correlated (p=0.0007, r=−0.4133) with lower alcohol DID intakes (*Figure 1o*). Together, this may suggest that the elevated inter-session DID water intakes correlate with decreased levels of alcohol intake during DID sessions for females and may offer another explanation for why females and males binge drank similar levels of alcohol in this behavioral paradigm.

## Sex-dependent differences in water and alcohol drinking mechanics

We also aimed to understand how well animals' intakes correlated with both water and alcohol-drinking microstructure measurements made directly from the lickometer in a sex-dependent manner. First, we plotted drinking events, drinking event duration, and number of drinks (i.e. cumulative drinking events that occurred in a 3- s window) and correlated them to water intake across all DID water sessions.

For water events, event duration, and number of drinks, were all significantly correlated with intake and there were no differences in how male and female mice consumed water out of the lickometers across all DID water sessions (*Figure 2—figure supplement 1a–c*). Similarly, alcohol events, event duration, and number of drinks were all highly correlated to alcohol intake (*Figure 2—figure supplement 1d–f*). For drinking events, there were no sex differences, yet for the alcohol event duration and number of alcohol drinks, female mice showed an increased correlation with alcohol intake at lower event numbers and drinks compared to males, suggesting they might be more efficient drinkers, such

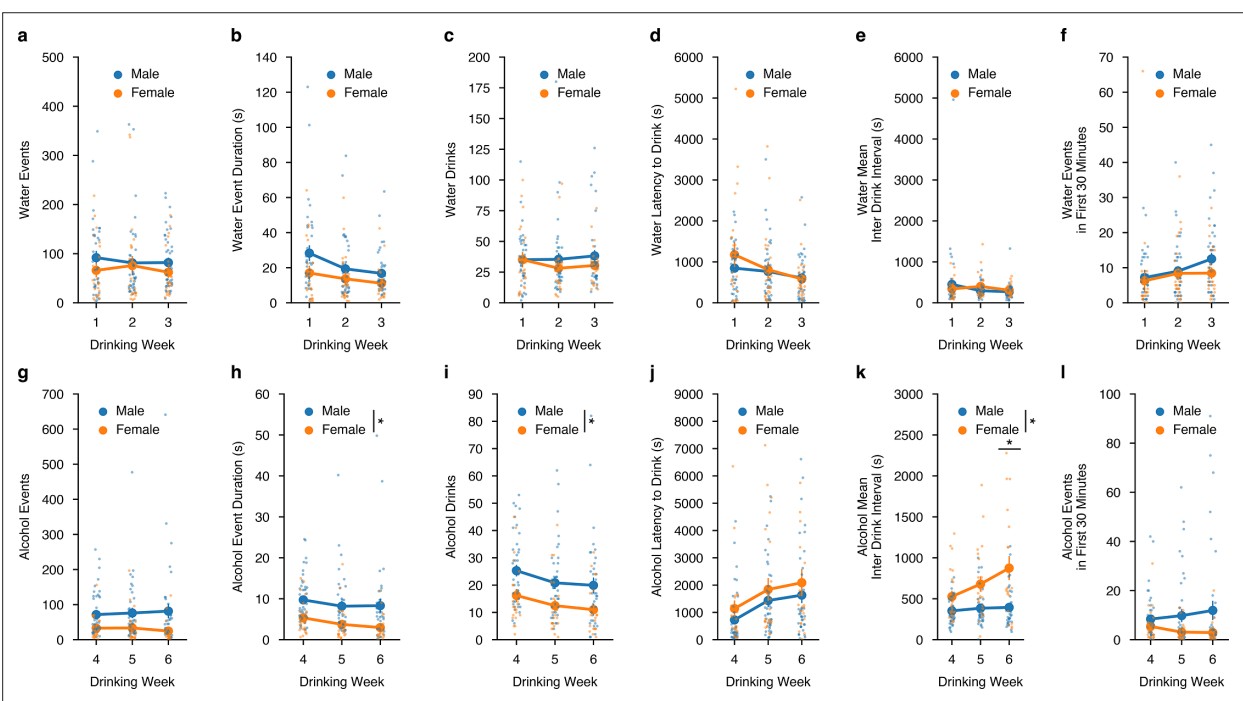

**Figure 2.** Alterations in drinking microstructure represent different strategies used to consume similar levels of alcohol, but not water intake in a sex-dependent manner. Water (**a**) events (mixed ANOVA, Sex $F_{(1,11)}$ = 0.6164, p=0.4490; male n=9, female n=5), (**b**) total event duration (mixed ANOVA, Sex $F_{(1,11)}$ = 1.422, p=0.2582; male n=9, female n=5), (**c**) drinks (mixed ANOVA, Sex $F_{(1,11)}$ = 0.3957, p=0.5422; male n=9, female n=5), (**d**) latency to drink (mixed ANOVA, Sex $F_{(1,11)}$ = 0.3911, p=0.5746; male n=9, female n=5) (**e**), mean inter-drink interval (mixed ANOVA, Sex $F_{(1,11)}$ = 0.0339, p=0.8572; male n=9, female n=5), and (**f**) events in the first 30 min of a DID session (mixed ANOVA, Sex $F_{(1,11)}$ = 0.538, p=0.4786; male n=9, female n=5) do not differ by sex across drinking week. Alcohol (**g**) events do not differ by sex across the week (mixed ANOVA, Sex $F_{(1,12)}$ = 3.183, p=0.0997; male n=9, female n=5), but alcohol (**h**) event duration (mixed ANOVA, Sex $F_{(1,12)}$ = 7.286, p=0.0193, np²=0.3778, Power (1 - β)=1.0; male n=9, female n=5), and (**i**) drinks are decreased in female compared to male binge alcohol drinkers (mixed ANOVA, Sex $F_{(1,12)}$ = 6.681, p=0.0239, np²=0.3576, Power (1 - β)=1.0; male n=9, female n=5). There were no sex differences in alcohol (**j**) latency to drink (mixed ANOVA, Sex $F_{(1,12)}$ = 1.914, p=0.1917; male n=9, female n=5) (**k**), but alcohol mean inter-drink interval is increased in females compared to males (mixed ANOVA, Sex x Week $F_{(2,24)}$ = 3.623, p=0.0422, np²=0.2319, Power (1 - β)=1.0; Week 6$t_{(6.81)}$=3.623, p=0.01882; male n=9, female n=5). There are no sex-dependent changes in (**l**) events in the first 30 min of an alcohol DID session by week (mixed ANOVA, Sex $F_{(1,12)}$ = 1.972, p=0.1856; male n=9, female n=5).

The online version of this article includes the following figure supplement(s) for figure 2:

**Figure supplement 1.** Microstructure features measured from lickometers correlate with alcohol and water intakes in a sex-dependent manner.

that efficiency measures the ability to consume more alcohol in a single drinking interaction compared to males (*Figure 2—figure supplement 1f*). Additionally, female mice may also be more exact in titrating their alcohol intakes to achieve the desired dose of alcohol compared to males, which could also be viewed as more efficient behavior.

We also wanted to see how these microstructure analyses were altered across time between sex. For water intake, across DID weeks 1–3, there were no sex differences in the number of events, event durations, water events, the latency to drink water, the mean inter-drink interval between water events, nor differences in front loading behavior for water intake (*Figure 2a–f*). For alcohol intake, across DID weeks 4–6 there were no sex differences for the number of alcohol events, but females did have a decreased time for alcohol event durations and number of alcohol drinks compared to males (*Figure 2g–i*). These findings across time further support that females may be more efficient in consuming alcohol, such that it requires less time and drinks to reach similar levels of alcohol intake that males achieve. Females and males displayed the same latency to drink alcohol across time, but as DID weeks progressed, females displayed an increased inter-drink interval (*Figure 2j–k*). Finally, both males and females displayed similar levels of front-loading behavior for alcohol (*Figure 2l*). Together, these data suggest that when alcohol intake levels are similar between males and females, the way in which they achieve those intakes may be dissimilar in both the manner in which females consume alcohol and how those differences in consumption evolve across time.

## Calcium dynamics of AIC inputs into the DLS during drinking bouts

To visualize changes in calcium dynamics at the AIC presynaptic inputs to both the left and right DLS, we aligned the changes in calcium fluorescence to the beginning of water and alcohol drinking events, creating a 10- s window with 5 se before the initiation of the drinking event and 5 s after.

First, to summarize the differences between calcium activity at left and right inputs to the DLS, we averaged all z-scored water and alcohol drinking sessions' calcium activity and compared them for sex, fluid, lateralization, and time differences. Altogether, it showed that drinking behaviors are largely encoded by male AIC inputs into the left DLS, with alcohol displaying the largest change in magnitude for ΔF/F both across time (*Figure 3a*). We also show robust differences in AIC signals that are sex, fluid, and lateralization-dependent across time during the drink window (*Figure 3a*). When analyzing the peak amplitude of the calcium signals, we also find that alcohol-left male AIC inputs show the largest magnitude, and water-left AIC inputs the second largest when time locked to drinking events (*Figure 3b*). Since AIC inputs into the DLS display strong lateralized effects, we split the data by left and right AIC inputs into the DLS to zoom in on the sex and fluid comparisons occurring at each input. By looking at all AIC inputs into the left DLS by sex and fluid, calcium activity is increased for alcohol compared to water across time for both sexes, but especially in males (*Figure 3c*). More detailed analyses of the dynamics of the calcium signals also displayed both sex and fluid differences for the peak amplitude of the calcium signal where males show increased peak amplitudes for both water and alcohol compared to females (*Figure 3d*). We also measured the time to the maximum calcium signal, and time to peak, during the drinking event to determine if alcohol or sex displayed a time-dependent shift in the on-kinetics of the calcium signal. Interestingly, a sex-by-fluid interaction shows that for males, binge alcohol consumption, compared to water, increases the amount of time that it takes for the peak calcium signal to occur (*Figure 3e*). Whereas in females, binge alcohol consumption, compared to water, decreases the time to peak (*Figure 3e*). Finally, we measured the area under the curve using ΔF/F values from the time when the peak ΔF/F value occurred to the time when the end of the drink occurred to assess the decay kinetics of the calcium signal. For left AIC inputs, the AUC measurements also showed both sex and fluid differences, such that the AUC for both water and alcohol measures were increased compared to females, suggesting that the calcium signals were larger and remained elevated for longer after the peak occurred (*Figure 3f*).

For AIC inputs into the right DLS by sex and fluid across time, compared to the left AIC inputs, these findings are reversed, such that a decrease in calcium for alcohol compared to water is present, especially in males, and that the sex differences, regardless of the fluid consumed, are mainly driven by differences in the off-kinetics of the calcium dynamics. (*Figure 3g*). In more detail, an interaction between sex and fluid for peak amplitude shows that males decrease their peak amplitude calcium signals as they transition to binge alcohol consumption (*Figure 3h*). For time to peak, like left AIC inputs, females see a decrease in time to peak as they switch from water to binge alcohol drinking,

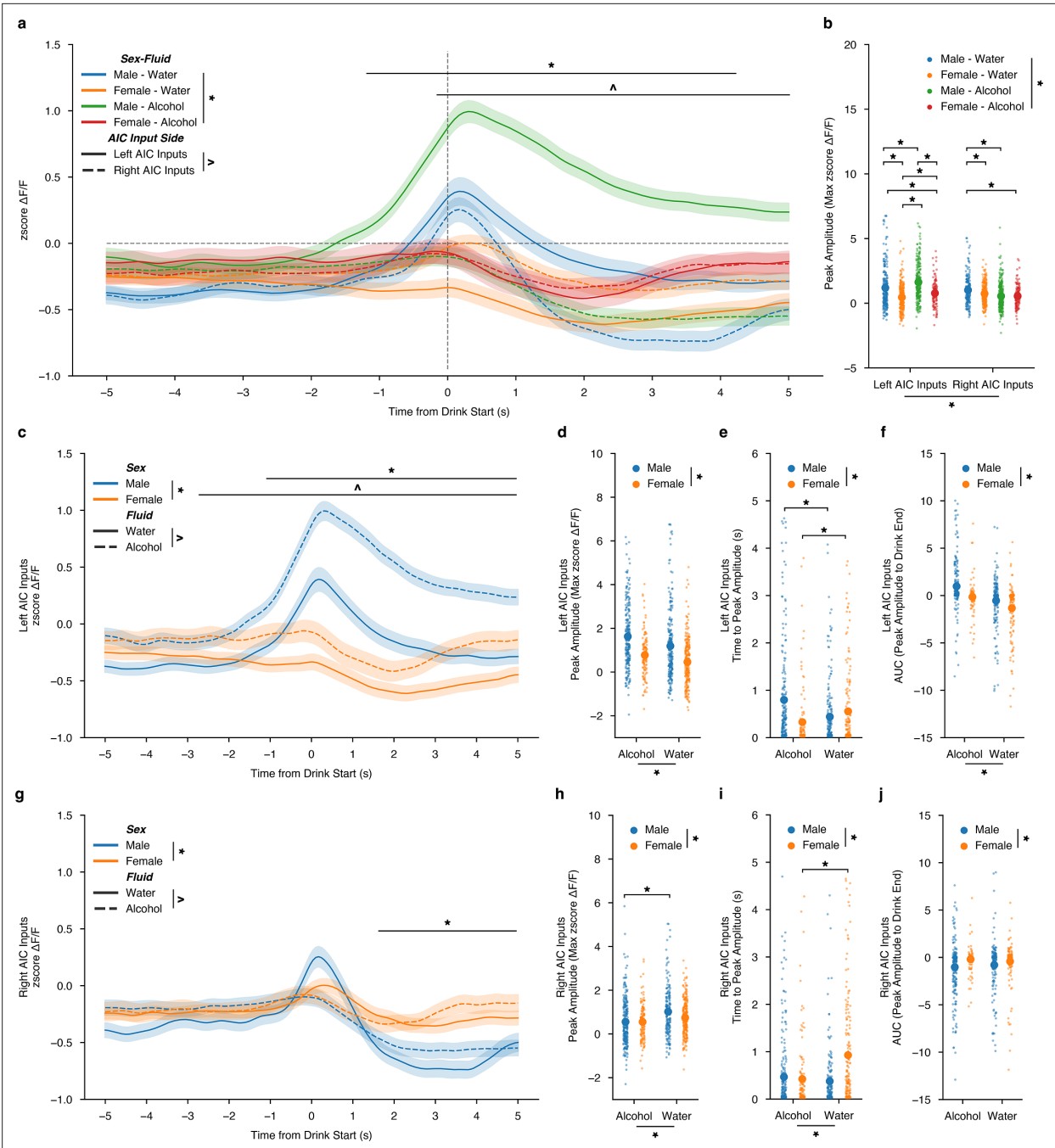

**Figure 3.** Sex-dependent calcium dynamics of anterior insular cortex (AIC) inputs into the left and right dorsolateral striatum (DLS) during water and drinking bouts. (**a**) AIC inputs to the DLS are most strongly engaged by alcohol at AIC inputs into the left DLS (ANOVA, Sex × Fluid × Time × Input Side $F_{(149, 200.7)}$=1.383, p=0.001383, $np^2$=0.000786, Power (1 - β)=1.0; *=Time from Drink Start × AIC Input Side, –0.0369128 (s) – 5 (s), p<0.05; ^=Time from Drink Start × Sex-Fluid, –1.308725 (s) – 4.95733 (s), p<0.05) and (**b**) peak amplitudes ΔF/F changes are strongly associated with changes in AIC inputs to the left DLS than with the right DLS (ANOVA, Sex x Fluid x Input Side $F_{(3,122.6)}$=31.15, p=1.298e-19, $np^2$=0.05083, Power (1 - β)=1.0). (**c**) For left AIC inputs into the DLS, there were both sex and fluid interactions by time for changes in ΔF/F (ANOVA, Sex × Fluid × Time $F_{(149,221.9)}$=1.404, p=8.419e-4, $np^2$=0.05083, Power (1 - β)=1.0; *=Time from Drink Start × Sex, –1.107383 (s) – 5 (s), p<0.05; ^=Time from Drink Start × Fluid, –2.181208 (s) – 5 (s), p<0.05) and (**d**) increases in peak amplitude for ΔF/F for alcohol compared to water were observed, but no interaction was present between sex and fluid (ANOVA, Fluid $F_{(1,32.45)}$=19.46, p=1.152e-5; Sex $F_{(1,132.2)}$=79.27, p=2.919e-18, $np^2$=0.05083, Power (1 - β)=1.0). (**e**) Time to peak values for males increased due to alcohol exposure, yet the opposite occurred for females (ANOVA, Sex × Fluid $F_{(1,14.25)}$=19.44, p=0.000012, $np^2$=0.02688, Power (1 - β)=0.9929; paired t-test, males $t_{(375.5)}$=4.0864, p=0.000107; paired t-test, females, $t_{(280.5)}$=–2.532, p=0.02361). (**f**) Left AUC were increased in males in comparison to females, but no interaction was present between sex and fluid (ANOVA, Fluid $F_{(1,332.3)}$=56.58, p=1.652e-13; Sex $F_{(1,149.8)}$=25.51, p=5.613e-7, $np^2$=0.074388, Power (1

*Figure 3 continued on next page*

*Figure 3 continued*

- β)=1.0). (**g**) For right AIC inputs into the DLS, there were both sex by time and fluid by time for changes in ΔF/F, but no three-way interaction (ANOVA, Sex × Time $F_{(149,772.6)}$=5.881, p=6.559e-103, np²=0.006891, Power (1 - β)=1.0; ^=Time from Drink Start × Sex, –1.778523 (s) – 5 (s), p<0.05; Fluid × Time $F_{(149,340.7)}$=2.594, p=5.977e-23, np²=0.003051, Power (1 - β)=1.0). (**h**) Decreases in peak amplitude ΔF/F for alcohol compared to water were observed for males, but not females (ANOVA, Sex × Fluid $F_{(1,3.652)}$=3.925, p=0.04787, np²=0.004641, Power (1 - β)=0.5096; paired t-test, males $t_{(407.1)}$=4.490, p=0.000009; paired t-test, females, $t_{(271.9)}$=–2.248, p=0.05015). (**i**) For females, time to peak decreased when transitioning from water to alcohol exposure, but there was no effect in males (ANOVA, Sex × Fluid $F_{(1,15.02)}$=20.99, p=0.000005, np²=0.02852, Power (1 - β)=0.9957; paired t-test, males $t_{(410)}$ = 1.288, p=0.1985; paired t-test, females, $t_{(289.6)}$ = - 4.558, p=0.000008). (**j**) For AUC, there was only a main effect of sex, with females displaying larger AUCs across both water and alcohol compared to males (ANOVA, Sex $F_{(1,61.39)}$=11.41, p = 0.00077, np²=0.01571, Power (1 - β)=0.9226). Genetically encoded calcium indicator (GCaMP) activity is aligned to the initiation of the water drinking bout with 5 s before and 5 s after the bout plotted. Solid lines represent means, and shading or error bars represent the standard error of the mean.

The online version of this article includes the following figure supplement(s) for figure 3:

**Figure supplement 1.** Calcium dynamics of anterior insular cortex (AIC) inputs into the left and right dorsolateral striatum (DLS) during water drinking.

**Figure supplement 2.** Left anterior insular cortex (AIC) inputs into the dorsolateral striatum (DLS) encode binge alcohol drinking in males, but not females.

yet there is no change in males (*Figure 3i*). Finally, there was only a sex-dependent decrease in males compared to females for AUC, regardless of fluid (*Figure 3j*). In summary, two main findings are apparent, such that (1) for left AIC inputs, regardless of sex, alcohol increases calcium activity compared to water during drinking events and for right AIC inputs, alcohol decreases calcium activity compared to water drinking events, and (2) that male calcium dynamics at both left and right AIC inputs are more strongly altered by alcohol exposure than female calcium dynamics. Together, we also observed a net shift towards increased engagement of AIC inputs into the left DLS during the transition from water to binge alcohol drinking in males, but not females, which is achieved via both increasing AIC engagement to the left DLS and decreasing AIC engagement to the right DLS during drinking events.

Since we uncovered fluid effects for both left and right AIC inputs across the time from drink start, we wanted to further examine the sex effects observed and to understand how AIC inputs evolved across the drinking task within each fluid. To do this, we filtered the dataset presented in *Figure 3* to include only water-drinking events and then analyzed for sex differences. Looking at all water events, we failed to detect a sex difference for both left and right AIC inputs into the DLS across the time from drink start (*Figure 3—figure supplement 1a,e*). For left AIC inputs during water drinking, we did see an increase in peak amplitude, no change in time to peak amplitude, and an increase in AUC for males compared to females (*Figure 3—figure supplement 1b–d*). For right AIC inputs, similarly, an increase in peak amplitude was observed (*Figure 3—figure supplement 1f*). Females also displayed an increased time to peak, but there was no sex effect for AUC (*Figure 3—figure supplement 1g–h*).

Additionally, observing how the development of water drinking, across weeks, altered calcium dynamics we found that for water drinking in males, there were no differences by DID week in either the left or right AIC inputs across time from drink starts (*Figure 4a,e*). For left inputs in water-drinking males, we observed an increase in the peak amplitude in week 3 compared to weeks 1 and 2, but no change in time to peak or AUC measurements (*Figure 4b–d*). For the right inputs in water-drinking males, we saw the opposite for peak amplitudes, a decrease in week 3 compared to week 1, no change in time to peak, and a decrease in the AUC in week 3 compared to week 1 (*Figure 4f–h*). For females, there were no differences by DID week in either the left or right AIC inputs across time from drink starts (*Figure 4j,m*). We see similar changes, such that left AIC inputs for water-drinking females also increase peak amplitude, with no change in time to peak by week, like left AIC inputs in water-drinking males (*Figure 4j–k*). Although, increases in AUC are seen in week 3 compared to week 1 (*Figure 4l*). For right AIC inputs, we see an increase in peak amplitude in week 3 compared to week 1, with a decrease in time to peak, and no changes in AUC (*Figure 4n–p*). Together, we see no effect across water drinking weeks when looking at the calcium signal in the entire water drinking window, and only small alterations in calcium signal when looking at specific parts (peak amplitude, time to peak, and AUC) of the calcium dynamics across sex. Generally, as the animals experience more water DID sessions, AIC inputs become more engaged during water drinking events as measured by increases in peak amplitude, with corresponding increases in AUC in some cases.

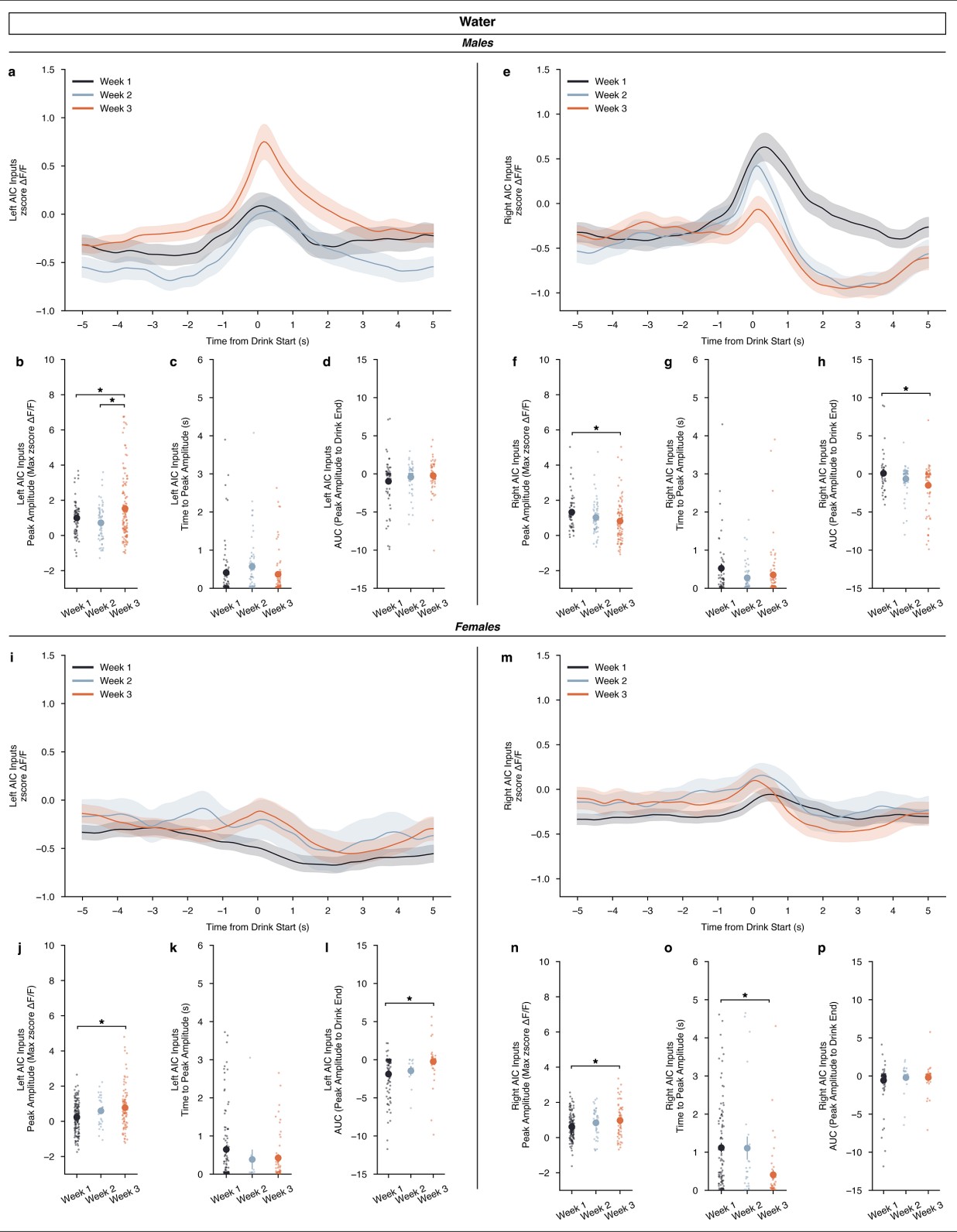

**Figure 4.** Calcium dynamics of anterior insular cortex (AIC) inputs into the left and right dorsolateral striatum (DLS) across water drinking weeks. (**a**) For males, left AIC inputs to the DLS do not display a by week-dependent change in ΔF/F (rmANOVA, DID Week × Time from Drink $F_{(298,298)}$ = 0.6953, p=0.5575), but (**b**) peak amplitudes increased in the final week of water drinking compared to the first two (ANOVA, DID Week $F_{(2,163.8)}$=7.235, p=0.000975, np²=0.05062, Power (1 - β)=0.9999; Week 3 – Week 1, p=0.02478; Week 3 – Week 2, p=0.000557). Yet, no difference in (**c**) time to peak

*Figure 4 continued on next page*

*Figure 4 continued*

(ANOVA, drinking in the dark (DID) Week $F_{(2,127.4)}$=1.556, p=0.215) or (**d**) AUC were observed by week (ANOVA, DID Week $F_{(2,129.9)}$=1.405, p=0.2491) for male, left AIC inputs. (**l**) Right AIC inputs to the DLS did also not display a by-week dependent change in ΔF/F (rmANOVA, DID Week × Time from Drink $F_{(298,596)}$ = 2.166, p=0.2743), but (**f**) a decrease in peak amplitudes in the final water drinking week was observed in comparison to the first week (ANOVA, DID Week $F_{(2,192)}$ = 3.5774, p=0.02983, np$^2$=0.03593, Power (1 - β)=0.9978; Week 3 – Week 1, p=0.02206). (**g**) No change in time to peak was observed by week (ANOVA, DID Week $F_{(2,104.6)}$=2.139, p=0.1229), but (**h**) a decrease in AUC was observed with the final week of water drinking displaying a deceased off-kinetic of the calcium signal in comparison to the initial drinking week (ANOVA, DID Week $F_{(2,113.9)}$=5.301, p=0.006291, np$^2$=0.06266, Power (1 - β)=0.9999; Week 3 – Week 1, p=0.004061). (**i**) For females, left AIC inputs to the DLS do not display a by week dependent change in ΔF/F (rmANOVA, DID Week × Time from Drink $F_{(298,298)}$ = 1.308, p=0.5817), but (**j**) like males, peak amplitudes increased in the final week of water drinking (ANOVA, DID Week $F_{(2,78.83)}$=7.529, p=0.001017, np$^2$=0.060465, Power (1 - β)=1.0; Week 3 – Week 1, p=0.00108). No difference in (**k**) time to peak (ANOVA, DID Week $F_{(2,30.37)}$=1.591, p=0.2203) was observed, but (**l**) AUC increased in the final week of water drinking compared to the initial week (ANOVA, DID Week $F_{(2,34.22)}$=7.0389, p=0.002751, np$^2$=0.07152, Power (1 - β)=0.9999; Week 3 – Week 1, p=0.000801). (**m**) Right AIC inputs to the DLS did also not display a by week dependent change in ΔF/F (rmANOVA, DID Week x Time from Drink $F_{(298,596)}$ = 1.022, p=0.4966), but (**n**) an increase in peak amplitudes in the final water drinking week was observed in comparison to the initial week (ANOVA, DID Week $F_{(2,93.65)}$=4.533, p=0.01321, np$^2$=0.03957, Power (1 - β)=0.9999; Week 3 – Week 1, p=0.01778). (**o**) A decline in time to peak was observed in the final week of water drinking (ANOVA, DID Week $F_{(2,58.75)}$=10.26, p=0.000151, np$^2$=0.07598, Power (1 - β)=0.9999; Week 3 – Week 1, p=0.00066), but (**p**) no change in AUC was observed by week (ANOVA, DID Week $F_{(64.13)}$=0.7386, p=0.4818). Genetically encoded calcium indicator (GCaMP) activity is aligned to the initiation of the water drinking bout with 5 s before and 5 s after the bout plotted. Solid lines represent means, and shading or error bars represent the standard error of the mean.

When performing the same analysis, i.e., filtering the dataset presented in *Figure 3*, but for only binge alcohol drinking sessions, we see that for left AIC inputs, we show a strong sex-dependent engagement from approximately the time to peak for about 2 s (*Figure 3—figure supplement 2a*). As a result, an increase in peak amplitude and AUC is also observed, with no change in time to peak across sex (*Figure 3—figure supplement 2b–d*). For right AIC inputs, we observed no change in calcium signal across time from drink start, no change in peak amplitude, nor any change in time to peak during binge alcohol consumption (*Figure 3—figure supplement 2e–g*), but we did observe a slight decrease in AUC that represents the decreased signal seen in the off-kinetics in males compared to females (*Figure 3—figure supplement 2h*). These data further detail how left AIC inputs strongly encode binge alcohol consumption in male, but not female mice.

Again, we also split the alcohol data by sex and looked at how the development of binge drinking, by week, altered calcium signal. For males, we did not find changes in calcium signal across time from drink start between weeks for either left or right AIC inputs (*Figure 5a & e*). For left AIC inputs in males, we did not find differences in peak amplitude, and time to peak, but we did find a decrease in the off-kinetics in week 6 compared to week 5 and week 4 (*Figure 5b–d*). This suggests that while alcohol strongly encodes drinking activity during binge alcohol consumption, that the calcium signal begins to decay quicker as the drinking history increases. For right AIC inputs, we observed an increase in week 5 compared to week 6 for peak amplitude, but no changes in time to peak or AUC (*Figure 5f–h*). For females, interestingly, we observed no changes in calcium signal by time for any of the metrics we analyzed (*Figure 5i–p*). By sex, males show strong engagement of left AIC signals and decreased right AIC signals across increasing binge alcohol history. Furthermore, the left AIC signal's off-kinetics as measured by AUC, decay in the final week of DID, suggesting that long-term exposure to binge alcohol consumption can alter the dynamics of left AIC calcium signals. Yet, for females, across the board, a lack of alcohol-induced changes over time is observed, which can be interpreted as a resistance or insensitivity to binge alcohol consumption across time.

Finally, we wanted to determine if acute water or binge alcohol consumption altered calcium dynamics within a single drinking session. For each DID session, we mapped the start times of drinks onto the full recordings of the left and right AIC input calcium signals. For an example DID session, see *Figure 6a–b*. Then, across all DID sessions, we plotted a histogram of all the drink lengths associated with each drinking event as well as the number of drinks that occurred within all sessions (*Figure 6c–d*). To compare short versus long drinks and early versus late drinks within the DID session, we calculated the quartile ranges of each measure and assigned those events to their respective range with pseudo colors for easy comparison.

We found no differences in calcium signals between sex or fluid as a function of different drink lengths (*Figure 6—figure supplement 1a–h*). For a number of drinks, we also found no differences in calcium signals between sex or fluid as a function of the number of drinks within a single session (*Figure 6—figure supplement 2a–h*). The pattern of calcium activity in male, left AIC inputs into

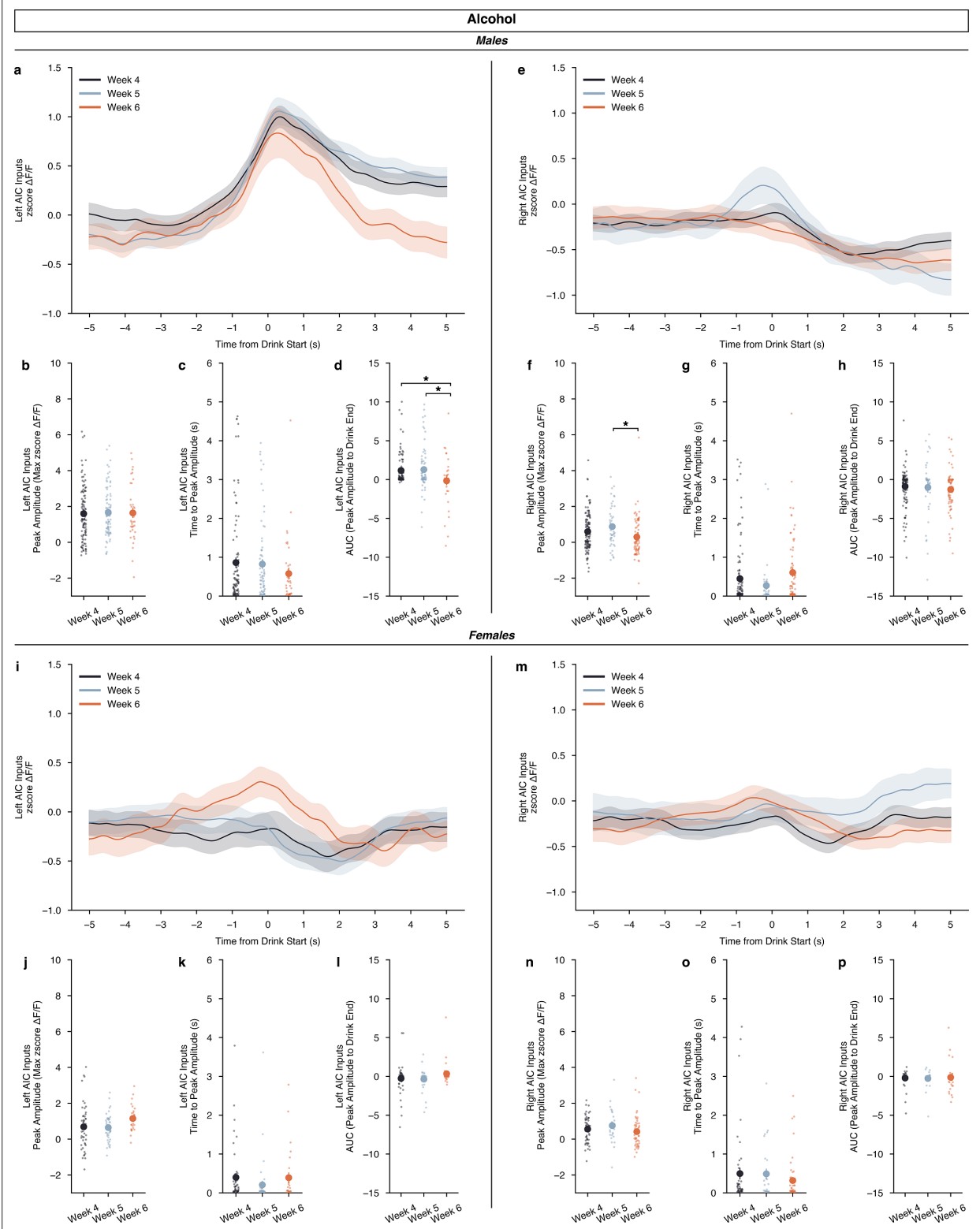

**Figure 5.** Calcium dynamics of anterior insular cortex (AIC) inputs into the left and right dorsolateral striatum (DLS) across binge alcohol drinking. (**a**) For males, left AIC inputs to the DLS do not display a week-dependent change in ΔF/F (rmANOVA, drinking in the dark (DID) Week × Time from Drink $F_{(298,298)} = 2.271$, p=0.3729), nor (**b**) did peak amplitudes change by binge week (ANOVA, DID Week $F_{(2,246)} = 0.05842$, p=0.9433). No difference in (**c**) time to peak were seen (ANOVA, DID Week $F_{(2,113.7)} = 1.359$, p=0.2609) but (**d**) decreases in AUC were observed in the final binge week compared to the first two (ANOVA, DID Week $F_{(2,91.76)} = 3.481$, p=0.03491, np²=0.04435, Power (1 - β)=0.9998: Week 4 – Week 6, p=0.04111; Week 5 – Week 6, p=0.03777) for

*Figure 5 continued on next page*

*Figure 5 continued*

male, left AIC inputs. (**e**) Right AIC inputs to the DLS did also not display a by week dependent change in ΔF/F (rmANOVA, DID Week × Time from Drink $F_{(298,596)}$ = 2.166, p=0.2743), but (**f**) an increase in peak amplitudes in the second week of binge drinking week was observed in comparison to the final week (ANOVA, DID Week $F_{(2,256)}$ = 4.8388, p=0.008655, np$^2$=0.03643, Power (1 - β)=0.9996; Week 5 – Week 6, p=0.007429). (**g**) No change in time to peak was observed by week (ANOVA, DID Week $F_{(2,109.8)}$=2.933, p=0.05744) (**h**), and no change in AUC was observed across binge drinking weeks for male, right AIC inputs into the DLS (ANOVA, DID Week $F_{(2,87.91)}$=0.5946, p=0.5539). (**i**) For females, left AIC inputs to the DLS do not display a by-week dependent change in ΔF/F (rmANOVA, DID Week × Time from Drink $F_{(298,596)}$ = 1.544, p=0.3716). (**j**) No change in peak amplitudes, (ANOVA, DID Week $F_{(2,128)}$ = 2.961, p=0.05534) (**k**) time to peak, (ANOVA, DID Week $F_{(2,70.21)}$=1.238, p=0.2962) or (**l**) AUC were observed (ANOVA, DID Week $F_{(2,69.62)}$=2.089, P=0.1315). (**m**) Right AIC inputs to the DLS did also not display a by week dependent change in ΔF/F (rmANOVA, DID Week x Time from Drink $F_{(298,596)}$ = 0.8997, p=0.4538). Similar to female, left AIC inputs, there were no changes in (**n**) peak amplitudes, (ANOVA, DID Week $F_{(2,135)}$ = 1.606, p=0.2045) (**o**) time to peak, (ANOVA, DID Week $F_{(2,70.11)}$=1.067, p=0.3495), or (**p**) AUC by binge drinking week either (ANOVA, DID Week $F_{(2,73.17)}$=0. 1216, p=0.8857). Genetically encoded calcium indicator (GCaMP) activity is aligned to the initiation of the water drinking bout with 5 s before and 5 s after the bout plotted. Solid lines represent means, and shading or error bars represent the standard error of the mean.

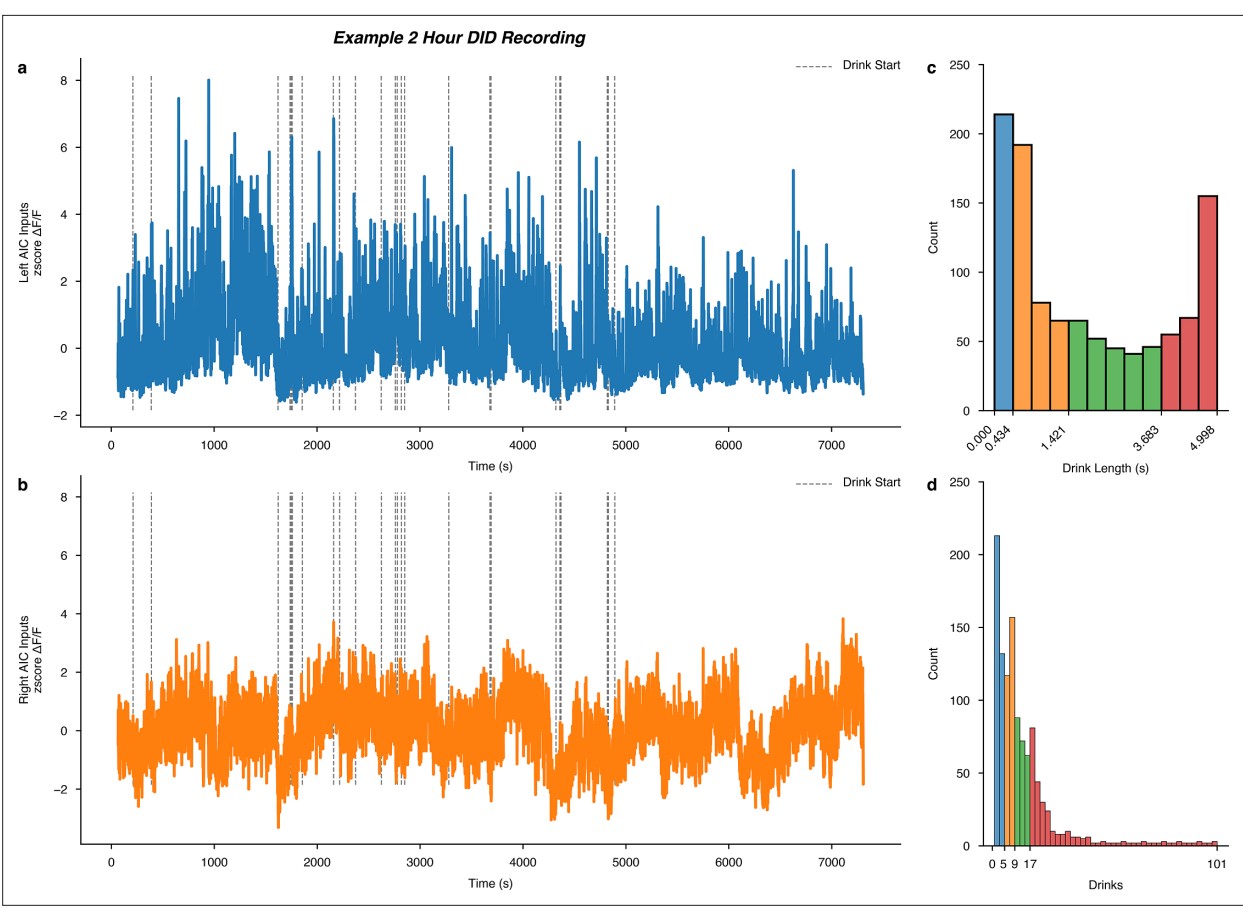

**Figure 6.** Intra-drinking in the dark (DID) session calcium signals and global quantiles of drink length and total number of drinks per session. Example male (**a**) left and (**b**) right anterior insular cortex (AIC) input to dorsolateral striatum (DLS) calcium recordings from a single 2 hr binge alcohol drinking in the dark (DID) recording (animal 'P7,' session 17), with dashed vertical lines indicating the start of an alcohol drinking event overlayed on the time-resolved genetically encoded calcium indicator (GCaMP) signal. A histogram of all measured (**c**) drink lengths and (**d**) the number of drinks observed across all water and alcohol drinking sessions. On the x-axis, the quartile ranges are printed and colored to visualize the quartile ranges of drink length and total number of drinks that occurred across all water and alcohol DID sessions.

The online version of this article includes the following figure supplement(s) for figure 6:

**Figure supplement 1.** Intra-drinking in the dark (DID) session calcium signals do not differ by the duration of each drinking event.

**Figure supplement 2.** Intra-drinking in the dark (DID) session calcium signals do not differ by the number of drinking events within a single session.

the DLS during binge alcohol drinking is striking, such that it may be the case that increasing acute intoxication decreases the calcium signal seen at these inputs within single DID sessions. However, we did not find statistical evidence of this in our dataset, but this may be because of how we un-biasedly chose to quartile the number of drinks observed and or increase alcohol tolerance as the number of binge sessions increases for any individual animals' alcohol history. For the purpose and scope of this work, we thought it was important to quantify some basic drinking metrics within DID sessions to ensure our weekly and between-sex and fluid analyses of calcium dynamics were in large part not driven by acute, intra-session effects. Ultimately, we found this to be true, but future work must be done to look at the acute effects of alcohol on AIC inputs within single DID sessions to follow up on these observations.

## Discussion

In summary, here we show how male and female mice can binge consume both water and alcohol in similar amounts, but via different behavioral strategies. Based on these data, we show how female mice may be more efficient alcohol drinkers, more susceptible to inter-session water drinking that modulates future alcohol intake, and time-resolved differences in how their binge drinking behaviors evolve compared to male mice. These robust measurements and resultant sex differences of binge alcohol-associated behavioral strategies must continue to be explored, but we hope these data serve as proof of the diversity between sexes that exist in alcohol consumption models even when animals consume similar amounts of alcohol.

The calcium imaging at pre-synaptic AIC inputs to the DLS data presented broadly suggest: (1) there is a time-dependent activation of AIC inputs into the DLS that occurs across drinking weeks where calcium activity, depending on the sex and lateralization, becomes more or less engaged to alcohol drinking and (2) lateralization affects insula projections into the DLS, such that insula signals received in the right and left DLS are remarkably different.

First, we saw strong evidence that AIC inputs into the DLS become engaged around water drinking behaviors, but ultimately most strongly engaged at AIC inputs into the left DLS early in binge alcohol consumption in male mice (*Figure 3c*). Interestingly, these robust calcium signals encoding binge consumption also decayed more rapidly as binge alcohol exposure history increased, suggesting that insular signals into the DLS can dynamically respond to alcohol across time (*Figure 5d*). Together, these data suggest that while AIC inputs into the left DLS are most strongly engaged during alcohol drinking, that the change in their kinetics and thus what they might be encoding in a behavioral and/or interoceptive context may be altered across time due to increasing, repetitive alcohol consumption. And, while these generalized dynamic changes in calcium activity are interesting observations, we did not find strong statistical evidence for changes in calcium signals by week within both water and alcohol drinking across the entire drinking window alone. Thus, while we did find changes in isolated measures such as peak amplitude, time to peak, and AUC by drinking week within water and alcohol drinking, future studies employing faster and higher resolution imaging techniques at cellular resolution, in combination with behavioral paradigms that provide more power at smaller time scales will be needed to understand these changes in calcium dynamics fully. Furthermore, we only measured calcium activity during water and alcohol drinking at AIC inputs into the DLS, but additional work will be required to tease apart changes to glutamate, GABA, and other specific in-vivo presynaptic neurotransmitter release in this circuitry as alcohol-induced neuroadaptations form, but nonetheless these data represent intriguing initial findings.

Second, lateralization of insular activity is well documented in human and rat studies, yet few, if any, mouse studies show clear lateralization effects within the insular cortex (*Centanni et al., 2021*). To our knowledge, this is the first demonstration, in a water or alcohol-drinking paradigm, to show lateralized effects of AIC inputs in the mouse. Also, in our previous work, we did not see lateralization effects when stimulating AIC inputs into the left DLS compared to the right DLS, but it is important to note that the time-resolution in which we were photo activating AIC inputs in the DLS was around 10 orders of magnitudes longer than the time-resolution of the measured calcium activity, suggesting we could have been occluding lateralization effects by stimulating for too long, at a non-biologically relevant frequency, and/or with too much light power (*Haggerty et al., 2022*).

Together, these findings suggest there exist both sex-dependent, fluid-dependent, and brain lateralization effects for AIC inputs into the DLS in how they encode water and alcohol drinking activity.

Here, we mainly focused on describing increases in calcium activity and its associated dynamics to demonstrate how male, left AIC inputs into the DLS are more robustly engaged during binge drinking behaviors. But, we also reported many instances where calcium signals were decreased, where shifts in time to peaks occurred, and where sustained decreases in calcium signals occurred for many seconds post-drinking initiation, especially on right AIC inputs in both males and females. It is known that these shifts, pauses, and decreases in firing can also have profound behavioral effects, and likely interact with the increases in calcium activity described to direct a concert of action selection behaviors relevant for both alcohol and water drinking behaviors. Again, work utilizing higher resolution in-vivo imaging and behavioral assays and analyses will be required to resolve these changes, but nonetheless, these observations provide initial evidence for a powerful, complex cortico-striatal circuit that has vast implications in addiction-related behaviors.

Furthermore, the findings presented here help contextualize our previous work on AIC input control over binge alcohol drinking behaviors such that alcohol exposure in males decreases robust calcium activity from AIC inputs that when restored with photoactivation of AIC inputs into the DLS leads to decreases in binge alcohol consumption (*Figure 5d*). Yet, in females, the lack of perturbed calcium dynamics may correlate with the lack of alcohol-induced changes in synaptic plasticity that were observed in slice electrophysiology at AIC inputs to the DLS after 3 weeks of binge alcohol exposure (*Haggerty et al., 2022*). In conclusion, leveraging these sex-dependent differences in AIC circuitry as binge drinking behaviors develop may represent novel biological levers to target in the quest for developing new pharmacotherapies and therapeutic modalities to treat those experiencing AUD.

## Materials and methods

### Animals

Animal care and experimental protocols for this study were approved by the Institutional Animal Care and Use Committee at the Indiana University School of Medicine and all guidelines for ethical protocols and care of experimental animals established by the National Institutes of Health (Bethesda, MD) were followed. Male and Female C57BL/6 J mice were ordered from the Jackson Laboratory (Bar Harbor, ME). Animals arrived at 6 weeks of age and were allowed to acclimate to the animal facility for at least 1 week before any surgery was performed. All animals were single-housed in a standard 12 hr light/dark cycle (lights on at 08:00 hr). Humidity in the vivarium was held constant at 50% and food and water were available ad libitum.

### Stereotaxic surgeries

All surgeries were conducted under aseptic conditions using an ultra-precise small animal stereotaxic instrument (David Kopf Instruments, Tujunga, CA). Animals were anesthetized using isoflurane (3% for induction, 1.0–1.5% for maintenance at a flow rate of 25–35 ml/min of oxygen depending on body weight at the time of surgery). Viral injections were performed using a 33-gauge microinjection needle (Hamilton Company, Reno, NV). Animals were treated post-operatively for 72 hr with daily injections of carprofen (5 mg/kg) and topical lidocaine on the surgical incision. Animals were allowed to recover for at least 1 week before behavioral assays began. Animals were assigned to groups randomly and after surgery, animal IDs were re-coded to blind experimenters to viral expression status.

All mice were injected bilaterally with AAV9-syn-FLEX-jGCaMP7s-WPRE (Addgene, 104491-AAV9) in the AIC at coordinates A/P:+2.25, M/L:±2.25, D/V: −3.10 (100 nl/injection, 25 nl/min infusion rate) and AAVrg-Ef1a-mCherry-IRES-Cre (Addgene, 55632-AAVrg) in the DLS at coordinates A/P:+0.7, M/L:±2.4, D/V: −3.5 to express GCaMP in AIC inputs to the DLS in a cre-dependent fashion. Bilateral 200 μm fiber optic cannulas were also inserted into DLS at coordinates A/P:+0.7, M/L:±2.4, D/V: −3.5. The fiber optic cannulas were secured to the skull using OptiBond Universal etchant (Henry Schein, Melville, NY) followed by Tetric EvoFlow light-cured, nano-optimized flowable composite (Henry Schein, Melville, NY), and the skin was closed over the top of the probe using Vetbond tissue adhesive (3 M, Saint Paul, MN).

## Drinking in the dark

The DID paradigm was based on the original DID procedure with two modifications (*Rhodes et al., 2005*). First, there were four 2 hr DID sessions (Monday–Thursday) and one 4 hr DID session (Friday), and second, all DID sessions were performed out of a single bottle of water or alcohol (20% v/v in water) via lickometers (described below) inserted into the cage at the beginning of the DID session, which were removed at the end of the DID session. Mice had *ab libitum* access to their standard water bottles at all other times.

To summarize the performed procedures, after animals recovered from surgery, they were singly housed and allowed to acclimate in a reverse 12 hr dark/light cycle (lights off at 06:00 hr) for 1 week. Wire feeding racks were removed from the cage and animals were floor-fed to reduce damage to fiber optic probes protruding from their skulls. Three hours into the dark cycle (09:00 hr) animals were transferred to a behavior room, covered with blankets to avoid light exposure, and the homecage water bottle was removed from the cage. The lickometer was inserted into the cage after the homecage water bottle was weighed to calculate the inter-session water intake. Animals stayed in their own home cages for DID, but the lids were replaced with a clear cage top to allow for two down video recordings of DID sessions.

Following the completion of 2- or 4 hr access, the lickometers were removed and the fluid bottles were weighed immediately after the session. Grams per kg (g/kg) of water and alcohol were computed from the difference in bottle weight and the density of water or 20% alcohol in water. The original cage tops were replaced, and animals were covered and transported back to their animal facility housing room. Mice were weighed post-DID session on Monday, Wednesday, and Friday (Tuesday and Thursday weights were equivalent to Monday and Wednesday, respectively). For Saturday and Sunday, the mice had no access to the lickometers as no procedures were performed. This 1 week repeating DID cycles are referred to as 'Drinking Weeks.'

## In-vivo fiber photometry

Fiber photometry acquisition was performed with a Neurophotometrics fiber photometry system (FP3002, Neurophotometrics LTD, San Diego, CA). Briefly, this system utilized a pulsating 415 nm purple light LED as an isosbestic control and a 470 nm blue-light LED to excite GCaMP, each pulsating at 30 hertz with a power of 50 µW as measured out of the tip of each fiber optic cable. The fluorescence light path includes a dichroic mirror to pass emitted fluorescence to a complementary metal-oxide semiconductor camera (FLIR BlackFly, Goleta, CA) traveling through a dual branching 200 µm fiber optic patch cable (Doric Lenses, Québec, QC, Canada) allowing for the recording of isosbestic and GCaMP fluorescence signals from both the left and right AIC inputs to the DLS, simultaneously. Animals were plugged into patch cabling and then placed back in their home cage for 5 min before lickometers were inserted and DID sessions started. Fluorescence signals from the camera were captured with Bonsai (https://bonsai-rx.org/docs), which also integrated and captured lickometer data via an Arduino Uno.

The fiber photometry signal was processed using a custom Python pipeline (code available at: https://github.com/dlhagger/AIC_Fiber_Photometry, copy archived at *Haggerty, 2024*). In summary, we removed the first 300 frames from each recording and fit a biexponential decay function to the 415 nm isosbestic channel for each recording, in each brain region. That fit was then robustly linearly scaled using a RANdom Sample Consensus algorithm (RANSAC) to the 470 nm GCaMP emission data. Finally, the 470 nm GCaMP emission data was lowpass filtered at 6 Hz. Together these preprocessing steps control for photobleaching and motion artifacts in the GCaMP recording. The acquired 470 nm GCaMP data was then divided by the scaled 415 nm isosbestic control fit to compute a corrected observed fluorescence value, or ΔF/F. For each recording, and each brain region, the process was repeated, removing recordings with poor biexponential decay fits, excessive motion artifacts, and/or for recordings in which cabling disconnects or tangles occurred. In total, 820 recordings were acquired and 36 were excluded from the dataset for further analysis. Left and right AIC input ΔF/Fs were then z-scored by session for comparison of signals across fluids, sex, brain side lateralization, and time.

For peri-event analysis, the initiation of each drinking bout was defined by a 10- s sliding recording window including the 5 s before the bottle contact and 5 s post bottle contact. All z-scored ΔF/F values from both left and right sides were then aligned to the drink start in this window.

## Histology

Animals were anesthetized with isoflurane and trans-cardially perfused with 15 ml of ice-cold phosphate-buffered saline (PBS) followed by 25 ml of ice-cold 4% paraformaldehyde (PFA) in PBS. Animals were decapitated and the brain was extracted and placed in 4% PFA for 24 hr. Brains were then transferred to 30% sucrose solution until they sank after which they were sectioned on a vibratome at a section thickness of 50 μm. Brain sections were mounted in serial order on glass microscope slides and stained with 4',6-diamidino-2-phenylindole to visualize nuclei. Fluorescent images were captured on a BZ-X810 fluorescent microscope (Keyence, Itasca, IL) using 4x and 20x air objectives. Injection site, viral expression, and the location of optogenetic probe implantations were determined from matching images to the Reference Allen Mouse Brain Atlas. Animals that did not have a viral expression in the brain regions of interest and/or did not have fiber optic cannula probe placements within the brain region of interest as confirmed by histology were excluded from the study.

## Lickometers

All alcohol and water drinking experiments were performed using custom-built homecage fluid monitors (i.e. 'lickometers') that were constructed with the following modifications (*Godynyuk et al., 2019*). Liquid monitoring was constantly sampled as a function of the state (open vs. closed) of an infrared beam directly in front of the fluid bottle valve, and data was written to device memory every 3 s. This 3- s window ensured drinking activity data was reliably saved to the hardware. Therefore, any tube interaction within a 3- s window (defined as a drink) was recorded as the total number of beam breaks (defined as events) for the total duration that the beam was broken (defined as event duration) within that time window.

The lickometers can hold two bottles, although we only used one for each device during these experiments. We randomized the bottle side location across animals and found no effect on drinking, thus data are presented collapsed on the bottle side.

Lickometer data was cleaned by fitting a linear model comparing the number of events by the event duration within each bout for every DID session performed by each animal. Events that had a residual value of greater than or less than 3 from the model fit were removed. This cleaning procedure removed events that were due to slow leaks (a new tube was used if a leak was detected for the next DID session) or chews on the bottles (*Grecco et al., 2022*; *Haggerty et al., 2022*). Once raw lickometer data were cleaned, no additional outliers were removed when computing events, event durations, drinks, and/or any other microstructure elements.

For in-vivo fiber photometry experiments that utilized lickometers for time-resolving drinking behaviors with calcium imaging data, the lickometer's state changes triggered TTL outputs. 5 V TTL signals were read using digital I/O pins on an Arduino Uno board flashed with the firmata firmware. The Arduino Uno's clock was synchronized to the fiber photometry acquisition software, so timestamps could be aligned to photometry data during data analysis with precise time resolution.

## Microstructure feature analyses

Using events, event duration, drinks, and the timestamps when these events occurred during the DID session, we calculated other drinking features such as latency to drink (time to first bream break after session initiation) and mean inter-drink interval time per session (the mean time between drinks within each DID session). We also calculated features for events in the first 30 min of each DID session as a measure of front-loading behavior.

## Modeling and statistics

Sample sizes for all experiments were determined based on our previously published experimental findings for electrophysiology and in-vivo behavioral assays such as DID (*Muñoz et al., 2018*; *Haggerty et al., 2022*).

Data preprocessing and machine learning modeling utilized SciPy (*Virtanen et al., 2020*), Statsmodels (*Seabold and Perktold, 2010*), Scikit-learn (*Pedregosa et al., 2011*), and TensorFlow (*Abadi et al., 2016*). Statistical analyses were performed using pingouin (V0.5.4, *Vallat, 2018*). Data visualization used matplotlib *Hunter, 2007* and seaborn (*Waskom, 2021*) libraries. For time series and repeated measures, we used two-, three-, or four-way mixed or repeated measures ANOVAs where appropriate, which represented time (session or week) as the repeated, within-subject variable, and

the factor (fluid type, viral expression, bottle side, sex, etc.) as the between-subjects factor. All mixed and repeated measures ANOVAs matched data by subject in the modeling to account for inter-animal variability. All mixed ANOVA data were tested to see if variances between factors were equal and normal using the Levene test. For one-way non-normal ANOVAs, we used Welch ANOVAs. If there was a main effect for factor, we used pairwise t-tests to determine post hoc significance and p-values were Sidak corrected. For correlations between two variables, we tested multivariate normality using the Henze-Zirkler test. If samples were normal, we used a Pearson's correlation to report r and p-values. If samples failed normality testing, we used a Shepherd pi correlation that returned the Spearman correlation after removing bi-variate outliers. To test for group differences in correlations that were statistically significant, we used an analysis of covariance (ANCOVA) where the covariate factor was sex. For tests of two factors, we tested for sample normality across factors using the Shapiro-Wilk omnibus test. For normal samples, we used Welch two-tailed t-tests to correct for unequal sample sizes. For samples that failed normality testing, we used the Mann-Whitney U test. For significant statistical tests that generated F values, we reported partial eta squared and power values achieved. For tests that reported T values, we reported Cohen's *d* and power values achieved. Post-hoc power analyses were performed using G*Power 3.1 to ensure reported significance values were not biased type II errors that could result from over all animal numbers (9 males and 5 females). All significance thresholds were placed at $p<0.05$ and all data and model fits are shown as mean ± standard error (68% confidence interval) of the mean (SEM) and/or by scatter plot with means and error bars indicating the SEM.

---

## Additional information

### Funding

| Funder | Grant reference number | Author |
| --- | --- | --- |
| National Institute on Alcohol Abuse and Alcoholism | 5F31AA029297 | David L Haggerty |
| National Institute on Alcohol Abuse and Alcoholism | 5R01AA027214 | Brady K Atwood |

The funders had no role in study design, data collection and interpretation, or the decision to submit the work for publication.

### Author contributions

David L Haggerty, Conceptualization, Data curation, Software, Formal analysis, Funding acquisition, Investigation, Visualization, Methodology, Writing - original draft, Writing - review and editing; Brady K Atwood, Conceptualization, Supervision, Funding acquisition, Project administration, Writing - review and editing

### Author ORCIDs

David L Haggerty https://orcid.org/0000-0002-1455-2557
Brady K Atwood https://orcid.org/0000-0002-7441-2724

### Ethics

Animal care and experimental protocols for this study were approved by the Institutional Animal Care and Use Committee at the Indiana University School of Medicine (IACUC #: 19017) and all guidelines for ethical protocols and care of experimental animals established by the National Institutes of Health (Maryland, USA) were followed.

Reviewer #1 (Public review): https://doi.org/10.7554/eLife.96534.3.sa1
Reviewer #2 (Public review): https://doi.org/10.7554/eLife.96534.3.sa2
Reviewer #3 (Public review): https://doi.org/10.7554/eLife.96534.3.sa3
Author response https://doi.org/10.7554/eLife.96534.3.sa4

# Additional files

## Supplementary files
• MDAR checklist

## Data availability
Source data and analysis code to reproduce all figures are hosted at https://github.com/dlhagger/AIC_Fiber_Photometry (copy archived at *Haggerty, 2024*).

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
