## [Editor Report · eLife assessment]

This **valuable** manuscript describes evidence of sex differences in specific corticostriatal projections during alcohol consumption, and this is noteworthy given the increasing rates/levels of drinking in females and their liability for Alcohol Use disorder. The authors provide **solid** evidence of the lateralisation of the activity of the circuit, but other evidence is **incomplete**, particularly with regard to how the drinking measure relates to intoxication. There are some inconsistencies that make it difficult to reconcile the photometry and behavioral data. The findings would benefit from causal assessment in the future. The findings will be of interest to researchers investigating functional circuitry underlying alcohol-driven behaviors.

---

## [Referee Report · Reviewer #1 (Public review)]

Summary:

This paper uses a model of binge alcohol consumption in mice to examine how the behaviour and its control by a pathway between the anterior insular cortex (AIC) to the dorsolateral striatum (DLS) may differ between males and females. Photometry is used to measure the activity of AIC terminals in the DLS when animals are drinking and this activity seems to correspond to drink bouts in males but not females. The effects appear to be lateralized with inputs to the left DLS being of particular interest.

Strengths:

Increasing alcohol intake in females is of concern and the consequences for substance use disorder and brain health are not fully understood, so this is an area that needs further study. The attempt to link fine-grained drinking behaviour with neural activity has the potential to enrich our understanding of the neural basis of behaviour, beyond what can be gleaned from coarser measures of volumes consumed etc.

Weaknesses:

The introduction to the drinking in the dark (DID) paradigm is rather narrow in scope (starting line 47). This would be improved if the authors framed this in the context of other common intermittent access paradigms and gave due credit to important studies and authors that were responsible for the innovation in this area (particularly studies by Wise, 1973 and returned to popular use by Simms et al 2010 and related papers; e.g., Wise RA (1973). Voluntary ethanol intake in rats following exposure to ethanol on various schedules. Psychopharmacologia 29: 203-210; Simms, J., Bito-Onon, J., Chatterjee, S. et al. Long-Evans Rats Acquire Operant Self-Administration of 20% Ethanol Without Sucrose Fading. Neuropsychopharmacol 35, 1453-1463 (2010).) The original drinking in the dark demonstrations should also be referenced (Rhodes et al., 2005). Line 154 Theile & Navarro 2014 is a review and not the original demonstration.

When sex differences in alcohol intake are described, more care should be taken to be clear about whether this is in terms of volume (e.g. ml) or blood alcohol levels (BAC, or at least g/kg as a proxy measure). This distinction was often lost when lick responses were being considered. If licking is similar (assuming a single lick from a male and female brings in a similar volume?), this might mean males and females consume similar volumes, but females due to their smaller size would become more intoxicated so the implications of these details need far closer consideration. What is described as identical in one measure, is not in another.

While the authors have some previous data on the AIC to DLS pathway, there are many brain regions and pathways impacted by alcohol and so the focus on this one in particular was not strongly justified. Since photometry is really an observational method, it's important to note that no causal link between activity in the pathway and drinking has been established here.

It would be helpful if the authors could further explain whether their modified lickometers actually measure individual licks. While in some systems contact with the tongue closes a circuit which is recorded, the interruption of a photobeam was used here. It's not clear to me whether the nose close to the spout would be sufficient to interrupt that beam, or whether a tongue protrusion is required. This detail is important for understanding how the photometry data is linked to behaviour. The temporal resolution of the GCaMP signal is likely not good enough to capture individual links but I think more caution or detail in the discussion of the correspondence of these events is required.

Even if the pattern of drinking differs between males and females, the use of the word "strategy" implies a cognitive process that was never described or measured.

---

## [Referee Report · Reviewer #2 (Public review)]

Summary:

This study looks at sex differences in alcohol drinking behaviour in a well-validated model of binge drinking. They provide a comprehensive analysis of drinking behaviour within and between sessions for males and females, as well as looking at the calcium dynamics in neurons projecting from the anterior insula cortex to the dorsolateral striatum.

Strengths:

Examining specific sex differences in drinking behaviour is important. This research question is currently a major focus for preclinical researchers looking at substance use. Although we have made a lot of progress over the last few years, there is still a lot that is not understood about sex-differences in alcohol consumption and the clinical implications of this.

Identifying the lateralisation of activity is novel, and has fundamental importance for researchers investigating functional anatomy underlying alcohol-driven behaviour (and other reward-driven behaviours).

Weaknesses:

Very small and unequal sample sizes, especially females (9 males, 5 females). This is probably ok for the calcium imaging, especially with the G-power figures provided, however, I would be cautious with the outcomes of the drinking behaviour, which can be quite variable.

For female drinking behaviour, rather than this being labelled "more efficient", could this just be that female mice (being substantially smaller than male mice) just don't need to consume as much liquid to reach the same g/kg. In which case, the interpretation might not be so much that females are more efficient, as that mice are very good at titrating their intake to achieve the desired dose of alcohol.

---

## [Referee Report · Reviewer #3 (Public review)]

Summary:

In this manuscript by Haggerty and Atwood, the authors use a repeated binge drinking paradigm to assess how water and ethanol intake changes in male in female mice as well as measure changes in anterior insular cortex to dorsolateral striatum terminal activity using fiber photometry. They find that overall, males and females have similar overall water and ethanol intake, but females appear to be more efficient alcohol drinkers. Using fiber photometry, they show that the anterior insular cortex (AIC) to dorsolateral striatum projections (DLS) projections have sex, fluid, and lateralization differences. The male left circuit was most robust when aligned to ethanol drinking, and water was somewhat less robust. Male right, and female and left and right, had essentially no change in photometry activity. To some degree, the changes in terminal activity appear to be related to fluid exposure over time, as well as within-session differences in trial-by-trial intake. Overall, the authors provide an exhaustive analysis of the behavioral and photometric data, thus providing the scientific community with a rich information set to continue to study this interesting circuit. However, although the analysis is impressive, there are a few inconsistencies regarding specific measures (e.g., AUC, duration of licking) that do not quite fit together across analytic domains. This does not reduce the rigor of the work, but it does somewhat limit the interpretability of the data, at least within the scope of this single manuscript.

Strengths:

- The authors use high-resolution licking data to characterize ingestive behaviors.

- The authors account for a variety of important variables, such as fluid type, brain lateralization, and sex.

- The authors provide a nice discussion on how this data fits with other data, both from their laboratory and others'.

- The lateralization discovery is particularly novel.

Weaknesses:

- The volume of data and number of variables provided makes it difficult to find a cohesive link between data sets. This limits interpretability.

- The authors describe a clear sex difference in the photometry circuit activity. However, I am curious about whether female mice that drink more similarly to males (e.g., less efficiently?) also show increased activity in the left circuit, similar to males. Oppositely, do very efficient males show weaker calcium activity in the circuit? Ultimately, I am curious about how the circuit activity maps to the behaviors described in Figures 1 and 2.

- What does the change in water-drinking calcium imaging across time in males mean? Especially considering that alcohol-related signals do not seem to change much over time, I am not sure what it means to have water drinking change.

---

## [Author Response]

The following is the authors’ response to the previous reviews.

**Public Reviews:**

**Reviewer #1 (Public Review):**
Summary:This paper uses a model of binge alcohol consumption in mice to examine how the behaviour and its control by a pathway between the anterior insular cortex (AIC) to the dorsolateral striatum (DLS) may differ between males and females. Photometry is used to measure the activity of AIC terminals in the DLS when animals are drinking and this activity seems to correspond to drink bouts in males but not females. The effects appear to be lateralized with inputs to the left DLS being of particular interest.Strengths:Increasing alcohol intake in females is of concern and the consequences for substance use disorder and brain health are not fully understood, so this is an area that needs further study. The attempt to link fine-grained drinking behaviour with neural activity has the potential to enrich our understanding of the neural basis of behaviour, beyond what can be gleaned from coarser measures of volumes consumed etc.Weaknesses:The introduction to the drinking in the dark (DID) paradigm is rather narrow in scope (starting line 47). This would be improved if the authors framed this in the context of other common intermittent access paradigms and gave due credit to important studies and authors that were responsible for the innovation in this area (particularly studies by Wise, 1973 and returned to popular use by Simms et al 2010 and related papers; e.g., Wise RA (1973). Voluntary ethanol intake in rats following exposure to ethanol on various schedules. Psychopharmacologia 29: 203-210; Simms, J., Bito-Onon, J., Chatterjee, S. et al. Long-Evans Rats Acquire Operant Self-Administration of 20% Ethanol Without Sucrose Fading. Neuropsychopharmacol 35, 1453-1463 (2010).)

We appreciate the reviewer’s perspective on the history of the alcohol research field. There are hundreds of papers that could be cited regarding all the numerous different permutations of alcohol drinking paradigms. This study is an *eLife* “Research Advances” manuscript that is a direct follow-up study to a previously published study in *eLife* (Haggerty et al., 2022) that focused on the Drinking in the Dark model of binge alcohol drinking. This study must be considered in the context of that previous study (they are linked), and thus we feel that a comprehensive review of the literature is not appropriate for this study.

The original drinking in the dark demonstrations should also be referenced (Rhodes et al., 2005). Line 154 Theile & Navarro 2014 is a review and not the original demonstration.

This is a good recommendation. We have added this citation to Line 33 and changed Line 154.

When sex differences in alcohol intake are described, more care should be taken to be clear about whether this is in terms of volume (e.g. ml) or blood alcohol levels (BAC, or at least g/kg as a proxy measure). This distinction was often lost when lick responses were being considered. If licking is similar (assuming a single lick from a male and female brings in a similar volume?), this might mean males and females consume similar volumes, but females due to their smaller size would become more intoxicated so the implications of these details need far closer consideration. What is described as identical in one measure, is not in another.

As shown in Figure 1, all measures of intake are reported as g/kg for both water and alcohol to assess intakes across fluids that are controlled by body weights. We do not reference changes in fluid volume or BACs to compare differences in measured lickometry or photometric signals, except in one instance where we suggest that the total volume of water (ml) is greater than the total amount of alcohol (ml) consumed in DID sessions, but this applies generally to all animals, regardless of sex, across all the experimental procedures.

In Figure 2 – Figure Supplement 1 we show drinking microstructures across single DID sessions, and that males and females drink similarly, *but not identically*, when assessing drinking measures at the smallest timescale that we have the power to detect with the hardware we used for these experiments. Admittedly, the variability seen in these measures is certainly non-zero, and while we are tempted to assume that there exist at least some singular drinks that occur identically between males and females in the dataset that support the idea that females are simply just consuming more volume of fluid per singular drink, we don’t have the sampling resolution to support that claim statistically. Further, even if females did consume more volume per singular drink that males, we do not believe that is enough information to make the claim that such behavior leads to more “intoxication” in females compared males, as we know that alcohol behaviors, metabolism, and uptake/clearance all differ significantly by sex and are contributing factors towards defining an intoxication state. We’ve amended the manuscript to remove any language of referencing these drinking behaviors as identical to clear up the language.

No conclusions regarding the photometry results can be drawn based on the histology provided. Localization and quantification of viral expression are required at a minimum to verify the efficacy of the dual virus approach (the panel in Supplementary Figure 1 is very small and doesn't allow terminals to be seen, and there is no quantification). Whether these might differ by sex is also necessary before we can be confident about any sex differences in neural activity.

We provide hit maps of our fiber placements and viral injection centers, as we have, and many other investigators do regularly for publication based on histological verification. Figure 1A clearly shows the viral strategy taken to label AIC to DLS projections with GCaMP7s, and a representative image shows green GCaMP positive terminals below the fiber placement. Considering the experiments, animals without proper viral expression did not display or had very little GCaMP signal, which also serves as an additional expression-based control in addition to typical histology performed to confirm “hits”. These animals with poor expression or obvious misplacement of the fiber probes were removed as described in the methods. Further, we also report our calcium signals as z-scored differences in changes in observed fluorescence, thus we are comparing scaled averages of signals across sexes, and days, which helps minimize any differences between “low” or “high” viral transduction levels at the terminals, directly underneath the tips of the fibers.

While the authors have some previous data on the AIC to DLS pathway, there are many brain regions and pathways impacted by alcohol and so the focus on this one in particular was not strongly justified. Since photometry is really an observational method, it's important to note that no causal link between activity in the pathway and drinking has been established here.

As mentioned above, this article is an *eLife* Research Advances article that builds on our previous AIC to DLS work published in *eLife* (Haggerty et al., 2022). Considering that this is a linked article, a justification for why this brain pathway was chosen is superfluous. In addition, an exhaustive review of all the different brain regions and pathways that are affected by binge alcohol consumption to justify this pathway seems more appropriate to a review article than an article such as this.

We make no claims that photometric recordings are anything but observational, but we did observe these signals to be different when time-locked to the beginning of drinking behaviors. We describe this link between activity in the pathway and drinking throughout the manuscript. It is indeed correlational, but just because it is not causal does not mean that our findings are invalid or unimportant.

It would be helpful if the authors could further explain whether their modified lickometers actually measure individual licks. While in some systems contact with the tongue closes a circuit which is recorded, the interruption of a photobeam was used here. It's not clear to me whether the nose close to the spout would be sufficient to interrupt that beam, or whether a tongue protrusion is required. This detail is important for understanding how the photometry data is linked to behaviour. The temporal resolution of the GCaMP signal is likely not good enough to capture individual links but I think more caution or detail in the discussion of the correspondence of these events is required.

The lickometers do not capture individual licks, but a robust quantification of the information they capture is described in Godynyuk et al. 2019 and referenced in multiple other papers (Flanigan et al. 2023, Haggerty et al. 2022, Grecco et al. 2022, Holloway et al. 2023) where these lickometers have been used. However, individual lick tracking is not a requirement for tracking drinking behaviors more generally. The lickometers used clearly track when the animals are at the bottles, drinking fluids, and we have used the start of that lickometer signal to time-lock our photometry signals to drinking behaviors. We make no claims or have any data on how photometric signals may be altered on timescales of single licks. In regard to how AIC to DLS signals change on the second time scale when animals initiate drinking behaviors, we believe we explain these signals with caution and in context of the behaviors they aim to describe.

Even if the pattern of drinking differs between males and females, the use of the word "strategy" implies a cognitive process that was never described or measured.

We use the word strategy to describe a plan of action that is executed by some chunking of motor sequences that amounts to a behavioral event, in this case drinking a fluid. We do not mean to imply anything further than this by using this specific word.

**Reviewer #2 (Public Review):**
Summary:This study looks at sex differences in alcohol drinking behaviour in a well-validated model of binge drinking. They provide a comprehensive analysis of drinking behaviour within and between sessions for males and females, as well as looking at the calcium dynamics in neurons projecting from the anterior insula cortex to the dorsolateral striatum.Strengths:Examining specific sex differences in drinking behaviour is important. This research question is currently a major focus for preclinical researchers looking at substance use. Although we have made a lot of progress over the last few years, there is still a lot that is not understood about sex-differences in alcohol consumption and the clinical implications of this.Identifying the lateralisation of activity is novel, and has fundamental importance for researchers investigating functional anatomy underlying alcohol-driven behaviour (and other reward-driven behaviours).Weaknesses:Very small and unequal sample sizes, especially females (9 males, 5 females). This is probably ok for the calcium imaging, especially with the G-power figures provided, however, I would be cautious with the outcomes of the drinking behaviour, which can be quite variable.For female drinking behaviour, rather than this being labelled "more efficient", could this just be that female mice (being substantially smaller than male mice) just don't need to consume as much liquid to reach the same g/kg. In which case, the interpretation might not be so much that females are more efficient, as that mice are very good at titrating their intake to achieve the desired dose of alcohol.

We agree that the “more efficient” drinking language could be bolstered by additional discussion in the text, and thus have added this to the manuscript starting at line 440.

I may be mistaken, but is ANCOVA, with sex as the covariate, the appropriate way to test for sex differences? My understanding was that with an ANCOVA, the covariate is a continuous variable that you are controlling for, not looking for differences in. In that regard, given that sex is not continuous, can it be used as a covariate? I note that in the results, sex is defined as the "grouping variable" rather than the covariate. The analysis strategy should be clarified.

In lines 265-267, we explicitly state that the covariate factor was sex, which is mathematically correct based on the analyses we ran. We made an in-text error where we referred to sex as a grouping variable on Line 352, when it should have been the covariate. Thank you for the catch and we have corrected the manuscript.

But, to reiterate, we are attempting to determine if the regression fits by sex are significantly different, which would be reported as a significant covariate. Sex is certainly a categorical variable, but the two measures at which we are comparing them against are continuous, so we believe we have the validity to run an ANCOVA here.

**Reviewer #3 (Public Review):**
Summary:In this manuscript by Haggerty and Atwood, the authors use a repeated binge drinking paradigm to assess how water and ethanol intake changes in male in female mice as well as measure changes in anterior insular cortex to dorsolateral striatum terminal activity using fiber photometry. They find that overall, males and females have similar overall water and ethanol intake, but females appear to be more efficient alcohol drinkers. Using fiber photometry, they show that the anterior insular cortex (AIC) to dorsolateral striatum projections (DLS) projections have sex, fluid, and lateralization differences. The male left circuit was most robust when aligned to ethanol drinking, and water was somewhat less robust. Male right, and female and left and right, had essentially no change in photometry activity. To some degree, the changes in terminal activity appear to be related to fluid exposure over time, as well as within-session differences in trial-by-trial intake. Overall, the authors provide an exhaustive analysis of the behavioral and photometric data, thus providing the scientific community with a rich information set to continue to study this interesting circuit. However, although the analysis is impressive, there are a few inconsistencies regarding specific measures (e.g., AUC, duration of licking) that do not quite fit together across analytic domains. This does not reduce the rigor of the work, but it does somewhat limit the interpretability of the data, at least within the scope of this single manuscript.Strengths:- The authors use high-resolution licking data to characterize ingestive behaviors.- The authors account for a variety of important variables, such as fluid type, brain lateralization, and sex.- The authors provide a nice discussion on how this data fits with other data, both from their laboratory and others'.- The lateralization discovery is particularly novel.Weaknesses:- The volume of data and number of variables provided makes it difficult to find a cohesive link between data sets. This limits interpretability.

We agree there is a lot of data and variables within the study design, but also believe it is important to display the null and positive findings with each other to describe the changes we measured wholistically across water and alcohol drinking.

- The authors describe a clear sex difference in the photometry circuit activity. However, I am curious about whether female mice that drink more similarly to males (e.g., less efficiently?) also show increased activity in the left circuit, similar to males. Oppositely, do very efficient males show weaker calcium activity in the circuit? Ultimately, I am curious about how the circuit activity maps to the behaviors described in Figures 1 and 2.

In Figure 3C, we show that across the time window of drinking behaviors, that female mice who drink alcohol do have a higher baseline calcium activity compared to water drinking female mice, so we believe there are certainly alcohol induced changes in AIC to DLS within females, but there remains to be a lack of engagement (as measured by changes in amplitude) compared to males. So, when comparing consummatory patterns that are similar by sex, we still see the lack of calcium signaling near the drinking bouts, but small shifts in baseline activity that we aren’t truly powered to resolve (using an AUC or similar measurements for quantification) because the shifts are so small. Ultimately, we presume that the AIC to DLS inputs in females aren’t the primary node for encoding this behavior, and some recent work out of David Werner’s group (Towner et al. 2023) suggests that for males who drink, the AIC becomes a primary node of control, whereas in females, the PFC and ACC, are more engaged. Thus, the mapping of the circuit activity onto the drinking behaviors more generally represented in Figures 1 and 2 may be sexually dimorphic and further studies will be needed to resolve how females engage differential circuitry to encode ongoing binge drinking behaviors.

- What does the change in water-drinking calcium imaging across time in males mean? Especially considering that alcohol-related signals do not seem to change much over time, I am not sure what it means to have water drinking change.

The AIC seems to encode many physiologically relevant, interoceptive signals, and the water drinking in males was also puzzling to us as well. Currently, we think it may be both the animals becoming more efficient at drinking out of the lickometers in early weeks and may also be signaling changes due to thirst states of taste associated with the fluid. While this is speculation, we need to perform more in-depth studies to determine how thirst states or taste may modulate AIC to DLS inputs, but we believe that is beyond the scope of this current study.

**Recommendations for the authors:**

**Reviewer #1 (Recommendations For The Authors):**
Line 45 - states alcohol use rates are increasing in females across the past half-decade. I thought this trend was apparent over the past half-century? Please consider revising this.

According to NIAAA, the rates of alcohol consumption in females compares to males has been closing for about the past 100 years now, but only recently are those trends starting to reverse, where females are drinking similar amounts or more than males.

Placing more of the null findings into supplemental data would make the long paper more accessible to the reader.

In reference to reviewer’s three’s point as well, there is a lot of data we present, and we hope for others to use this data, both null and positive findings in their future work. As formatted on *eLife’s* website, we think it is important to place these findings in-line as well.

**Reviewer #2 (Recommendations For The Authors):**
In addition to the points raised about analysis and interpretation in the Public Review, I have a minor concern about the written content. I find the final sentence of the introduction "together these findings represent targets for future pharmacotherapies.." a bit unjustified and meaningless. The findings are important for a basic understanding of alcohol drinking behaviour, but it's unclear how pharmacotherapies could target lateralised aic inputs into dls.

There are on-going studies (CANON-Pilot Study, BRAVE Lab, Stanford) for targeted therapies that use technologies like TMS and focused ultrasound to activate the AIC to alleviate alcohol cravings and decrease heavy drinking days. The difficulty with these next-generation therapeutics is often targeting, and thus we think this work may be of use to those in the clinic to further develop these treatments. We agree that this data does not support the development of pharmacotherapies in a traditional sense, and thus have removed the word and added text to reference TMS and ultrasound approaches to bolster this statement in lines 101+.